# Research on Damage Properties of Apples Based on Static Compression Combined with the Finite Element Method

**DOI:** 10.3390/foods11131851

**Published:** 2022-06-23

**Authors:** Xiaopeng Liu, Zhentao Cao, Liu Yang, Huan Chen, Yonglin Zhang

**Affiliations:** 1School of Animal Science and Nutritional Engineering, Wuhan Polytechnic University, Wuhan 430048, China; lxp1989@whpu.edu.cn; 2College of Mechanical Engineering, Wuhan Polytechnic University, Wuhan 430048, China; jssysjmjczt@163.com (Z.C.); ch2293305261@163.com (H.C.); yin2482476406@163.com (Y.Z.); 3Hubei Cereals and Oils Machinery Engineering Center, Wuhan 430048, China

**Keywords:** apple, mechanical–structural properties, damage, finite element method

## Abstract

Apples are easily damaged during transportation due to extrusion and collision, resulting in structural damage and deterioration. To better understand apples’ mechanical–structural damage behavior, a texture analyzer platform combined with in situ observation was established. The effects of extrusion distance, speed, working temperature, and typical kinds of apple were considered for damage mechanisms. Apple damage was analyzed via the finite element method (FEM). The results indicated that the apple extrusion behavior can be divided into elastic interaction and plastic damage. Compression displacement effects were obviously significant in terms of structural damage, and apple samples were in an elastic stage with displacement of less than 2.3 mm, and no structural damage. The peak force energy-displacement mathematical model was established, showing an “s” shape and upward parabolic shape. The critical compression energy was around 100N·mm during elastic interaction. The damaged area was positively correlated with the compression energy. The FEM simulation results were consistent with the damage distribution of apples. The effects of speed on the three apple types were different. Red Fuji apples with a bruised area were not sensitive to pressure speed. The effect on the crack forming of Ralls apples was significant. Golden Delicious apples with a bruised area and crack formation showed an intermediate effect. The peak force–temperature fitting curve showed a downward parabolic shape and an R^2^ determination factor of 0.99982. Apple squeeze damage mechanisms provide theoretical guidance for apple damage control.

## 1. Introduction

Fruit is an important part of the human diet; it provides essential micronutrients, phenols, organic acids, antioxidants, vitamins, and dietary fiber [1,2,3,4], and fruit phytochemicals play an important role in reducing the risk of chronic diseases [5,6]. Apples are among the world’s most popular fruit species, with more than 80 million tons of apples produced annually worldwide. China ranks first in annual production (more than 40 million tons). Apples rank third among the furthest-transported fruits in the world [7,8,9]. However, before reaching consumers, apples can experience mechanical damage after picking from the orchard, transportation, reloading, and storage. Studies have shown that 50% of apple damage occurs during these periods [5,10,11]. Understanding the fruit’s deformation behavior under dynamic drop impact loading has become an important issue [9]. The mechanical–structural damage mechanisms of apples have thus become a research focus. This has obvious implications for fruit preservation, transportation, and packaging.

At present, researchers have carried out lots of research on apple damage—especially in the field of apple damage mechanism and detection research. Studies have shown that little bruising occurs in the pre-harvest stage, and damage at this stage is usually not easily controlled [12]. Apple damage generally manifests as tissue browning, mostly due to external impact loading [13,14,15,16,17]. Excessive surface pressure can cause damage to the fruit’s flesh tissue. Oxidation reactions occur when the liquid from the fruit’s flesh escapes. The magnitude of the force generated at the moment of impact can affect the color and location of the bruise formed on the apple. Researchers have found that impact testing is a commonly used destructive method [18,19,20]. Non-destructive tests are also commonly used [21]. However, these experimental methods cannot describe the internal bruising phenomenon in the case of dynamic deformation. Bruising is the destruction of subcutaneous tissue without rupture of the epidermis, resulting in unseen damage [22]. Finite element analysis offers a promising way to solve such complex loading conditions of fruits, as the simulation technique allows visualization of model changes in the research field of agricultural deformation and damage analysis [23,24,25]. The effects of temperature on apple quality were studied by Hun-Joong Kweonthe. The modulus of elasticity decreased with increasing fruit temperature, and temperature was positively correlated with bruise area [5,26]. The effect of drop height on apple quality was studied by Komarnicki [11,15,20].

Damage to tissue caused by mechanical interaction directly affects its storage and transportation. In this study, the mechanical–structural damage properties of apples were analyzed. To establish a platform for in situ observation and texture analysis, three varieties of apples—Red Fuji, Ralls, and Golden Delicious—were selected for mechanical–structural property testing of apples at maturity. The effects of pressing distance, pressure, speed, and temperature were analyzed, while the stress distribution and damage properties of apples were analyzed using a combination of finite element models to reveal the mechanical–structural damage mechanisms of apples.

## 2. Materials and Methods

### 2.1. Experimental Material and Test Rig

The three types of apple samples used in this study were harvested in 2021. All were purchased from the market—Red Fuji from Shaanxi, Golden Delicious from Shandong, and Ralls from Liaoning. Fifty apples with undamaged surfaces were selected from each of the three test varieties. The apples were 75–80 mm in diameter at the equator. They were then stored at room temperature (20 °C).

The damage properties of the apples were investigated using a self-developed texture analyzer. The principle schematic is shown in Figure 1a. The test system consisted of four main components: a vertical loading component, compression measurement, load measurement, and in situ observation component. We set the test parameters and then performed tests to analyze the damage properties.

### 2.2. Test Sample Preparation and Test Method for TPA with Different Parts of Apples

To research the mechanical properties of different parts of apples, the apples needed to be pretreated as standard samples. The preparation process was as follows: The fruits were selected uniformly, without damage, and cut transversely, as shown in Figure 1b_1_. Apple texture profile analysis (TPA) testing was performed immediately after cutting. The apples were screened under the same conditions. The apples were tested for crispness, stickiness, and hardness. The trigger force was set to 0.02 N, the pre-test and post-test speed were set to 2.0 mm/s, and the depth of downward pressure was set to 5.0 mm. A cylindrical probe with diameter 2 mm was selected. The test method selected was TPA.

### 2.3. Sample Preparation and Compression Testing

Compression testing was performed using a self-developed texture analyzer. A cylindrical indenter of 65 mm diameter was used for the test. Firstly, the carrier table position was adjusted. The cylindrical indenter’s center was aligned with the geometric center of the carrier table. The sensor and probe height were calibrated. The experimental objects and parameters were set as shown in Table 1. Research on the influence of temperature was carried out by holding apple specimens in an electrically heated oven for 1 h. In the experiment, warmed apple samples were separated longitudinally into halves with a knife, as shown in Figure 1b_2_. They were then removed for compression experiments. The prepared apple samples were placed on a load-bearing platform. A compression probe was calibrated by different orientations so that the probe was pressed on the apple sample at the same position for each test. The experiment was repeated five times. The external structural changes of the apples during compression were recorded with an in situ observation device. The test apple samples were left to stand overnight, and the apple samples’ damage map was collected by super-eye. A schematic diagram of the experimental setup and the observation positions is shown in Figure 1b.

### 2.4. Apple Model Development and FE Simulation Parameter Settings

Based on the actual 3D properties of the apples, the 3D and FE models are shown in Figure 2a,b. The models were imported into ANSYS software. The mechanical parameters of the apples were set as follows: Poisson’s ratio 0.35 and elasticity modulus 5.0 MPa [27]. The probe material was set to structural steel. The apple material was set as the new fruit flesh material. Apple model compression may result in partial plasticization and larger bending. The model can change significantly, and needs adjustment to nonlinear case calculation [28]. Considering the state change and geometric nonlinearity of the seed compression process, the structural elements with plasticity, large deflection, and large strain capacity were used in the seed model [29]. Differences in peel and flesh were ignored to simplify the model, by assuming that the materials were all homogeneous [27,30]. Subsequently, the apple and probe were meshed separately. Since the simulation mainly monitored the stresses in the apple, the meshing was detailed in order to obtain more data. The boundary conditions were set, the base plate was fixed, and the contact between the apple and the base plate was set as frictional contact. Load was defined as displacement. A longitudinal monitoring line was added inside the apple model; this line was located in the direction normal to the model and the plate used to observe the stress distribution at pressure displacements of 2, 4, 6, and 8 mm.

## 3. Results and Discussion

### 3.1. Structural and Distributive Mechanical Properties of Apples

TPA tests were conducted on apples at a temperature of 10 °C, pressing distance of 5 mm, and speed of 2 mm/s. The mechanical properties of different apple samples were investigated; the apples were cut along the equator, as shown in Figure 3a. The test points were selected with a depth interval of 5 mm in the direction indicated by the kernel of the apple pentagon. The curves were obtained as shown in Figure 3b, where the horizontal axis represents apple depth and the vertical axis represents hardness. The flesh hardness at a depth of 5 mm from the surface was 0.90248 N/mm^2^. The flesh hardness at a depth of 25 mm was 0.58272 N/mm^2^. The source of the hardness measurement method was compared with the method of Słupska et al. [31,32]. The mechanical structure of the apple was different from that of a uniformly textured rubber ball. The hardness of apples tended to increase from the inside to the outside.

### 3.2. Apple Compression Properties

In order to analyze the apple compression behavior, Red Fuji was selected to be tested for damage with in situ observation at a speed of 2 mm/s, 10 °C, and distance of 8 mm. The force–pressing distance curve is shown in Figure 4a. Figure 4b shows the damage morphology at distances of 2, 4, 6, and 8 mm. The apple sample was first deformed elastically in the initial compression stage. The initial curve part was nearly linear. At point a, the damage to the apple appeared on the internal surface position of the flesh. The pressure was 112.5 N at this point. Upon further compression into the a–b stage, the force growth rate was significantly greater than the elastic yielding stage. This was due to the increased compression distance, pressing more of the upper flesh tissue. In the b–c, stage deeper flesh was crushed and rupture damage occurred. There was more ruptured flesh at this time; therefore, multiple peaks appeared. After the compression proceeded to point c, significant decreases in the pressure growth rate could be seen. For this reason, large amounts of flesh broke down, and the hardness decreased significantly. The volume of ruptured flesh was larger than that of the undamaged flesh. This stage involves more compression of the ruptured tissue. After the compression in stages c–d, the apple skin could not regain its original shape due to the compression of the fruit flesh.

### 3.3. Effects of Pressing Distance on Apples’ Mechanical Properties

#### 3.3.1. Compression Test at Different Pressing Distances

Compression displacement directly leads to different degrees of apple deformation. It is fundamental to research the mechanical–structural properties of apples. Figure 5 shows the peak force–displacement and energy–displacement relationships for the three types of apple at compression distances of 1, 2, 4, 6, and 8 mm, a speed of 2 mm/s, and 10 °C. Peak force–displacement and energy–displacement relation curves are shown in Figure 5a,b, respectively. The curves were fitted using polynomial functions. The fitted mathematical model for peak force–pressing distance and energy–pressing distance were obtained as follows:(1)$$y=a_{1}+a_{2}x+a_{3}x^{2}+a_{4}x^{3}$$

The fitted parameters are shown in Table 2. The peak force–pressing distance fitting curve showed an “S” shape, and the energy–pressing distance fitting curve was a downward parabola. The three apples were in the elastic phase with the smallest energy slope at a compression distance of less than 2.3 mm. The pressure increased as a convex function of the pressing distance at a pressing distance of about 2.3–6 mm. A large amount of fleshy tissue broke at this time, and this stage also had the most waves. The pressure growth rate slowed down at a pressing distance of 6–8 mm. At this time, the wave crest was also significantly less than in the previous stage. As this stage had a relatively large percentage of ruptured flesh, the main focus at this point was on compacting the resulting ruptured flesh. The peak force relation of the three apple varieties at 8 mm was Red Fuji > Golden Delicious > Ralls. The peak force of Red Fuji was maximal at a compression distance of 8 mm. Due to of Red Fuji having the highest water content, the pulp compressed out of the water could not be discharged, and filled the interior of the apple. The resistance of the Red Fuji apples became bigger when loaded and compressed. The apple itself absorbed energy when pressed. The three types of apples tested began to suffer damage after absorbing energy greater than about 100 Nmm. The pressure growth rate slowed down at a pressure distance of 6–8 mm, but the energy required increased at a greater rate. The bruise area expanded linearly with increasing applied force, but it expanded nonlinearly with time.

#### 3.3.2. FE Results on Compression Tests

To study the relationship between apple damage area and stress distribution, static FE simulations of apple extrusion were conducted under different pressing distances. The simulation results are shown in Figure 6, where the Von Mises stress cloud reflects that the stress concentration area exists inside the apple. The effects of compression distance on the equivalent stress clouds were analyzed under 2, 4, 6, and 8 mm conditions, as shown in Figure 6a–d. From the stress distribution, the maximum stress was concentrated near the initial contact point between the apple and the compression probe, and the maximum stress decreased from near the contact point to the surrounding diffusion. Due to the regular shape of the apple, the stress distribution was symmetrical. The stress distribution formed a nearly circular area. The obtained stress distribution diagrams for static loads were similar to the impact load stress distribution diagrams of Celik et al. [24,28]. When the pressing distance increased to 4 mm, the stress concentration started to appear in the contact area between the apple and the base plate. In order to observe the internal stress distribution of the apple from the 3D space, the model was dissected, as shown in Figure 6e; at this time, the compression distance was 8 mm. The stress distribution was similar to the shape of the apple damage, in a funnel shape. Maximum stress was located normal to the contact surface of the apple and the probe. Figure 6f shows a scatterplot of the stress distribution on the black line in Figure 6e at different compression distances, where it can be seen that stress is concentrated at the maximum compression distance.

### 3.4. Effects of Speed on Apple Damage Properties

During transportation, apples are impacted at different speeds. Therefore, to investigate the relationship between speed and apple damage, three types of apples were compressed at speeds of 0.1, 1, 2, 4, and 8 mm/s, at a compression distance of 6 mm, and at 5 °C. The peak apple force–compression rate line graph was generated as shown in Figure 7. It can be seen that the effect of the compression rate on the peak force of Red Fuji apples was not significant, and the difference in peak force was within the range of 20 N. The peak forces of the three apples were the closest at a rate of 2 mm/s. The damage map of the apple specimens at a pressure speed of 2 mm/s was taken with the super-eyes, as shown in Figure 8. By comparison, it can be seen that the damage to Red Fuji apples was in the form of area bruising, that to Ralls apples was in the form of internal flesh cracking, and that to Golden Delicious apples was in the form of internal bruising combined with flesh cracking. Different forms of internal damage can cause different responses of peak force to velocity. The effect of velocity on peak force is significant in the case of damage in the form of cracks.

### 3.5. Effects of Temperature on the Damage Properties of Red Fuji Apples

Apples are widely consumed and available all year round. Different temperatures during transportation can have different impacts on apple damage. Therefore, apple damage tests were performed at temperatures of 10, 20, 30, 40 and 50 °C, at a compression distance of 6 mm and speed of 2 mm/s. The peak force–temperature curve was obtained as shown in Figure 9. It can be seen that the peak force of apples tended to decrease slowly with the increase in temperature. At a temperature of 10 °C, the peak apple force was about 317 N. At a temperature of 50 °C, the peak apple force was about 276 N. The peak force decreased slowly from 10 to 20 °C; when the temperature exceeded 20 ℃, the apple peak force decreased at a significantly faster rate. Fruit are softer at higher storage temperatures [33,34]. The experimental data were obtained by fitting the relationship between apple temperature and peak force using polynomials. The resulting polynomial is as follows:(2)$$F=a_{1}+a_{2}t+a_{3}t^{2}+a_{4}t^{3}$$
where each parameter is as follows: a_1_ is 316.2372, a_2_ is 0.61162, a_3_ is −0.04461, a_4_ is 3.199 × 10^−4^, and R^2^ is 0.99982.

## 4. Conclusions

Mechanical–structural damage behavior was researched in-depth in three typical types of apples, using a self-developed texture analyzer with in situ observation. The effects of extrusion distance, speed, working temperature, and typical kinds of apple were considered during extrusion. The damage distribution was analyzed with FEM. Celik et al. conducted impact experiments on apples to simulate damage during transport; however, static loading can also damage apples [9]. Apple crushing damage mechanisms provide theoretical guidance for apple damage control. The results of this research can be summarized as follows:The pressure distance has an obvious impact on the structural damage. When the extrusion strength was less than 2.3 mm, the apple was in the elastic stage, with no structural damage. A mathematical model of peak force, energy, and pressure distance was developed, showing the “s” shape and upward parabolic shape.Apples with different forms of damage were affected differently by pressure velocity. The tests showed that the speed had a significant effect on the peak force of Ralls apples damaged in the form of cracks; there was no significant effect on the peak force of Red Fuji apples to produce area damage.The peak force–temperature curve has a downward parabolic shape as the preservation temperature increases. The decrease in temperature for texture softening leads to an increase in the area of damage to the apples. When the temperature is too high, the odor of the apples changes.The equivalent effect force verifies that the apple damage is mainly diffused vertically to the interior.

## Figures and Tables

**Figure 1 foods-11-01851-f001:**
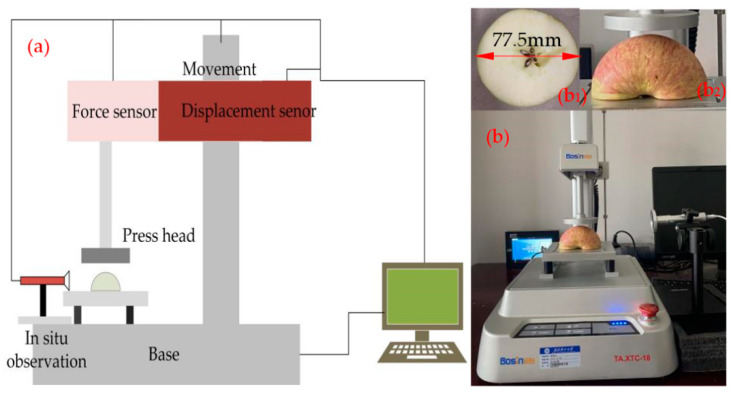
Schematic (**a**) and experimental setup (**b**) of the self-developed texture analyzer: (**b_1_**) TPA test sample; (**b_2_**) compressed test samples.

**Figure 2 foods-11-01851-f002:**
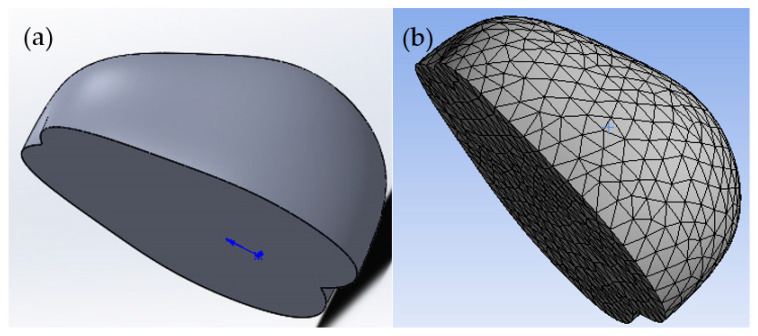
The 3D model (**a**) and FE mesh (**b**) of an apple sample.

**Figure 3 foods-11-01851-f003:**
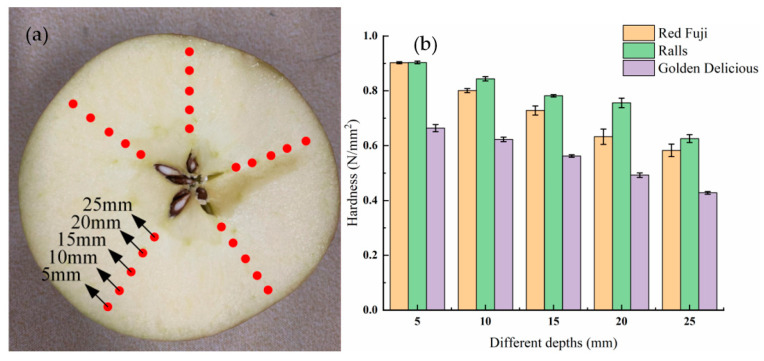
(**a**) Map of apple samples and test point locations; (**b**) bar chart of the hardness distribution of the three kinds of apple at different depths.

**Figure 4 foods-11-01851-f004:**
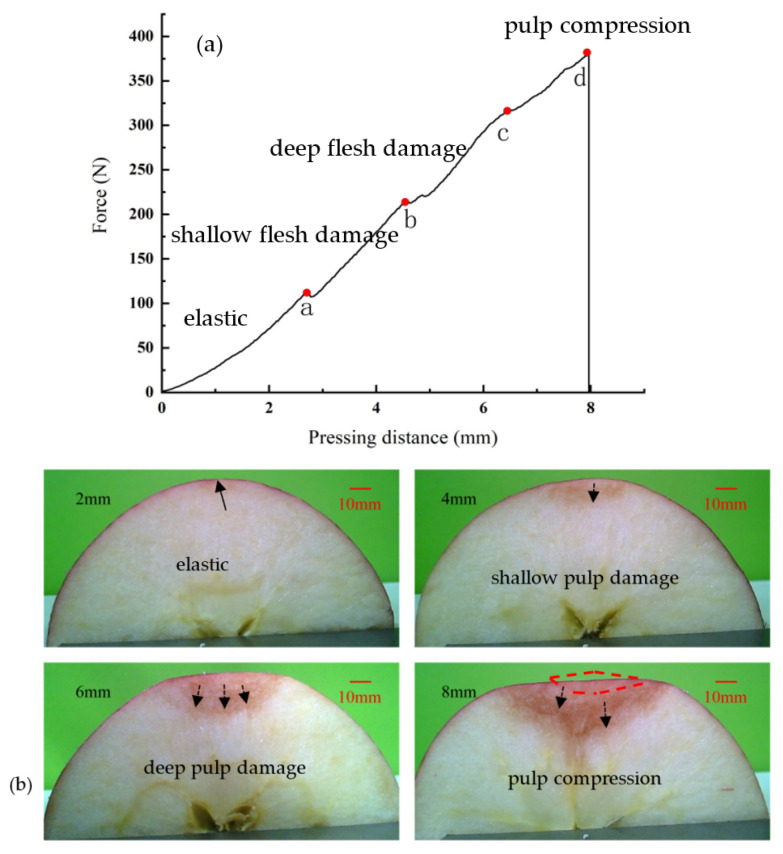
(**a**) Pressing distance–force curve at 8 mm distance and a speed of 2 mm/s—0–a: elasticity phase; a–b: shallow pulp damage stage; b–c: deep pulp damage stage; c–d: pulp compression stage. (**b**) Damage morphology under pressing at distances of 2, 4, 6, and 8 mm.

**Figure 5 foods-11-01851-f005:**
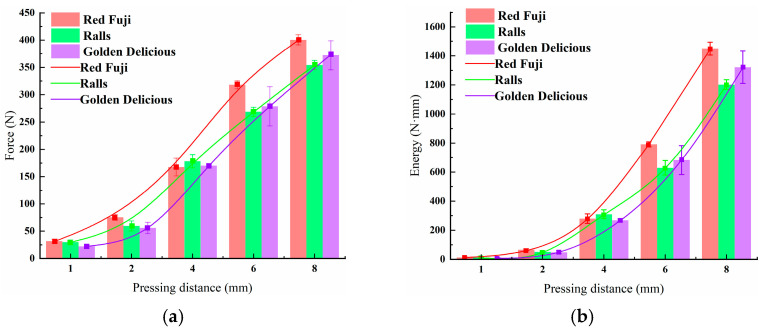
(**a**) Peak force–pressing distance and (**b**) energy–pressing distance fitting curves under a speed of 2 mm/s.

**Figure 6 foods-11-01851-f006:**
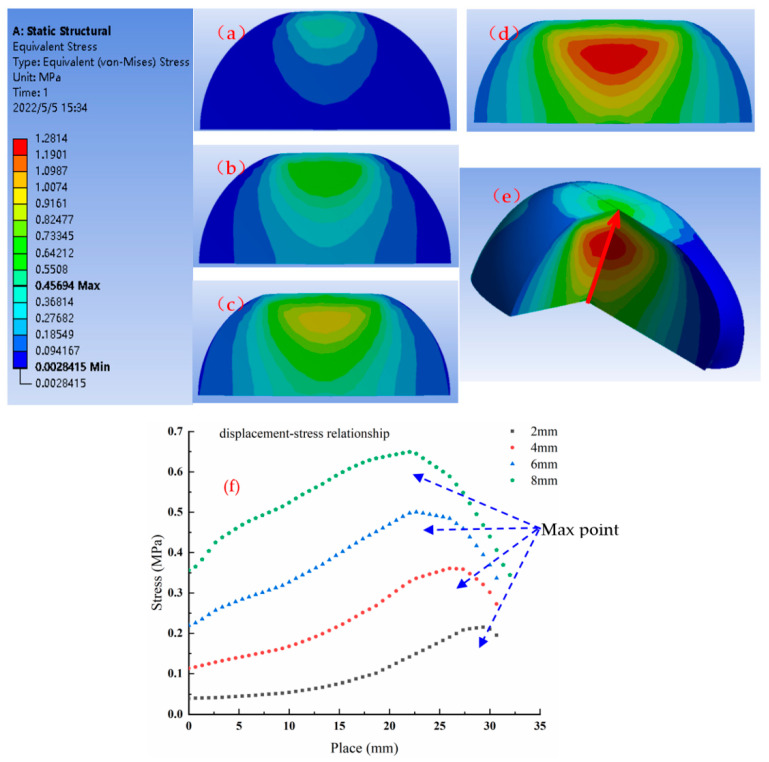
(**a**–**d**) Stress clouds dissected from the contact point between the apple and the plate parallel to the equatorial plane of the apple, at compression distances of 2, 4, 6, and 8 mm. (**e**) Added monitoring line in the pressure cloud of the apple model at a pressure distance of 8 mm. (**f**) Scatterplot of peak force at different locations and pressing distances. The zero-point *x*-axis is the bottom of the apple model.

**Figure 7 foods-11-01851-f007:**
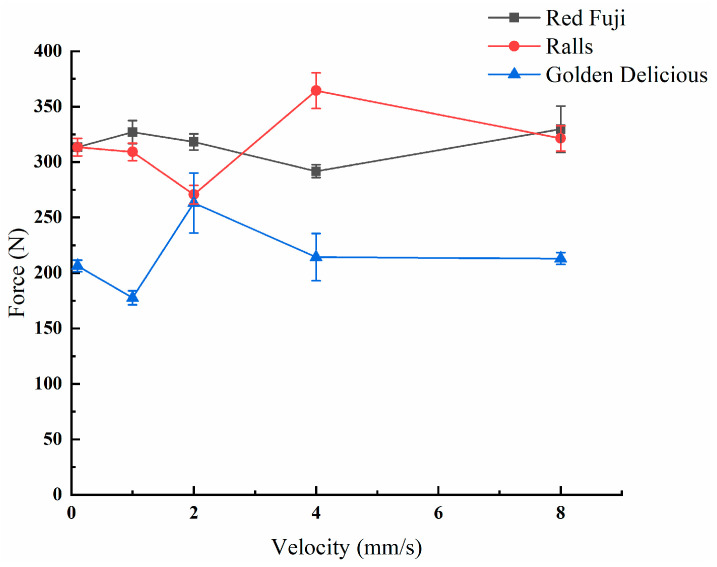
Peak force–speed relationships of different apple varieties, at a distance of 6 mm.

**Figure 8 foods-11-01851-f008:**
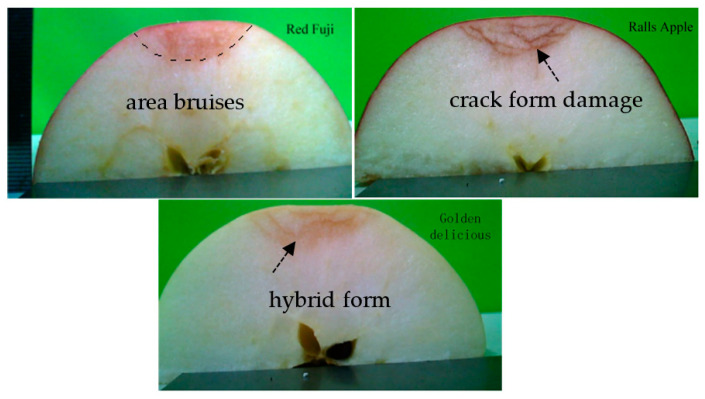
Damage morphology of different apple varieties under a pressing distance of 6 mm and a speed of 2 mm/s.

**Figure 9 foods-11-01851-f009:**
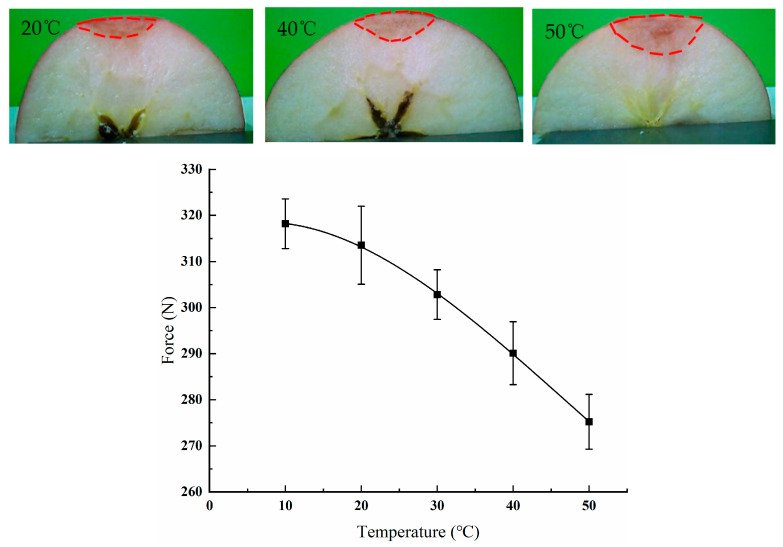
Peak force–temperature curve, and apple damage morphology at temperatures of 10, 20, 30, 40, and 50 °C.

**Table 1 foods-11-01851-t001:** Compression test parameters.

Variables	Species	Pressing Distance (mm)	Velocity (mm/s)	Temperature (°C)
1	Golden Delicious	1	0.1	10
2	Rad Fuji	2	1	20
3	Ralls	4	2	30
4	-	6	4	40
5	-	8	8	50

**Table 2 foods-11-01851-t002:** Peak force–pressing distance and energy–pressing distance fitting parameters for apples.

	a_1_	a_2_	a_3_	a_4_	R^2^
Red Fuji Peak Force	11.72965	8.79338	12.90509	−0.98859	0.99904
Red Fuji Energy	13.04673	−25.1784	25.25889	0.06092	0.99941
Ralls Peak Force	5.73311	15.67235	8.74748	−0.65923	0.99888
Ralls Energy	29.63372	−46.32897	30.65879	−0.82793	0.99882
Golden Delicious Peak Force	8.84371	−0.34938	14.64444	−1.11866	0.99982
Golden Delicious Energy	15.67289	−28.20406	21.80668	0.26254	0.99998

## Data Availability

The datasets generated and analyzed during the present study are available from the corresponding author upon reasonable request.

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
