# Peer review of "Research on Damage Properties of Apples Based on Static Compression Combined with the Finite Element Method"

_foods, 2022, doi:10.3390/foods11131851_

Round 1

Reviewer 1 Report

Questions to the authors: How was built the 3D apple model used for the FEM simulation ?

How many such 3D models have been used?

What software was used to analyze the image of the pictures of the smashed apples (Fig. 4)?

The axes in the diagrams should end with an arrows.

Reviewer 2 Report

The article 'Research on damage properties of apples based on static compression combined with FEM' gives a new approach to the knowledge of apple damage control. It's well written although it should be stated in conclusion how the results can be applied in practice.

Reviewer 3 Report

The topic of the research paper entitled ‘Research of damage properties of apples based on static compression combined with FEM’ fits well with the scope of Foods. The study investigates the mechanical damage behavior on 3 common apple varieties using a self-developed texture analyzer, in respect of extrusion distance, speed and temperature. The damage distribution was analyzed by Finite Element Method. Authors concluded that FEM results are consistent with damage distribution, the effects of speed are different for the 3 varieties studied and higher temperature leads to an increase of damaged area of the apples. Authors suggest that the results of the present study can provide guidance for apple damage control.

General comments:

               Although, the manuscript contains sufficient literature that covers the subject, it is poorly discussed in ‘Results and Discussion’ section. In detail, comparisons of the results of the current study with those of other similar are absent throughout section 3, as well as references to other published articles. There is only 1 citation in section 3, line 247, and the sentence needs to be rephrased.     

Manuscript:

1.      Title: authors are advised to avoid the use of abbreviations, such as FEM, in the title.

2.      Line 16: FEM, first time used, the abbreviation should be explained. Furthermore, the word ‘method’ after FEM can be avoided. From this point forward, authors have to decide whether they use FEM or FE method (e.g., line 261).   

3.      Line 33: ‘species’ instead of ‘varieties’.   

4.      Lines 57 and 60: references and their numbers have to be checked.

5.      Line 112: ‘fruit meat’ need to be rephrased.

6.      Line 122 – 124: sentences are confusing.

7.      Section 3.1: Were TPA tests conducted on all apple varieties? If yes, the hardness distribution chart (figure 3a) is the same for all varieties?

8.      Line 142: Why did you select Red Fuji? The other 2 varieties have a similar curve with the one represented in figure 4a?

9.      Lines 196, 198: delete ‘.’ after figure.

10.   Line 272: I believe it is ‘increase’ instead of ‘decrease’.

11.   Line 273: It is the first time that authors refer to ‘the odor of the apples’, if they believe it is an important aspect of the research, they can discuss it in Section 3.5.        

 Therefore, my recommendation is ‘Reconsider after major revision’.           

Round 2

Reviewer 3 Report

Authors have successfully addressed to all issues raised by the reviewers and the manuscript has been improved significantly. I, only, have the following 3 minor comments:         

1.      Line 124: replace ‘.’ with a ‘,’ after platen.

2.      Line 159: authors can add in the text that experiments were conducted on apples of all three kinds (data not shown), but since the overall diffusion form was similar for all 3 varieties, Red Fuji was selected for figure 4.  

3.      Line 162: add to the figure’s 4 legend, that these figures are for Red Fuji.

 Therefore, my recommendation is ‘Accept’.